# Bat sarbecovirus WIV1-CoV bears an adaptive mutation that alters spike dynamics and enhances ACE2 binding

Alexandra L. Tse[1], Gorka Lasso[1¤], Jacob Berrigan[1], Jason S. McLellan[2], Kartik Chandran [1]*, Emily Happy Miller[1,3]*

1 Department of Microbiology and Immunology, Albert Einstein College of Medicine, New York, New York, United States of America, 2 Department of Molecular Biosciences, The University of Texas at Austin, Austin, Texas, United States of America, 3 Department of Medicine, Albert Einstein College of Medicine, New York, New York, United States of America

¤ Current Address Division of Infection and Immunity, University College London, London, UK.
* kartik.chandran@einsteinmed.edu (KC); emily.miller@einsteinmed.edu (EHM)

## Abstract

SARS-like betacoronaviruses (sarbecoviruses) endemic in bats pose a significant zoonotic threat to humans. Genetic pathways associated with spillover of bat sarbecoviruses into humans are incompletely understood. We previously showed that the wild-type spike of the rhinolophid bat coronavirus SHC014-CoV has poor entry activity and uncovered two distinct genetic pathways outside the receptor-binding domain (RBD) that increased spike opening, ACE2 binding, and cell entry. Herein, we show that the widely studied bat sarbecovirus WIV1-CoV is likely a cell culture-adapted variant whose progenitor bears a spike resembling that of Rs3367-CoV, which was sequenced from the same population of rhinolophid bats as SHC014-CoV. Our findings suggest that the acquisition of a single amino-acid substitution in the '630-loop' of the S1 subunit was the key spike adaptation event during the successful isolation of WIV1-CoV, and that it enhances spike opening, virus-receptor recognition, and cell entry in much the same manner as the substitutions we previously identified in SHC014-CoV using a pseudotype system. The conformational constraints on both the SHC014-CoV and Rs3367-CoV spikes could be alleviated by pre-cleaving them with trypsin, suggesting that the spike-opening substitutions arose to circumvent the lack of S1–S2 cleavage. We propose that the 'locked-down' nature of these spikes and their requirement for S1–S2 cleavage to engage ACE2 represent viral optimizations for a fecal-oral lifestyle and immune evasion in their natural hosts. These adaptations may be a broader property of bat sarbecoviruses than currently recognized. The acquisition of a polybasic furin cleavage site at the S1–S2 boundary is accepted as a key viral adaptation for SARS-CoV-2 emergence that overcame a host protease barrier to viral entry in the mammalian respiratory tract. Our results suggest alternative spillover scenarios in which spike-opening substitutions that promote

which permits unrestricted use, distribution, and reproduction in any medium, provided the original author and source are credited.

**Data availability statement:** All data associated with this study, including all raw data required to replicate the results, are present in the paper, its Supplementary files, and the files shared through public repositories (see below). The raw data generated in this study has been deposited in the Figshare database under accession code https://doi.org/10.6084/m9.figshare.28779311. Source western blot images related to Figs 4A and S5A are available under accession code https://doi.org/10.6084/m9.figshare.28781000. Source western blot images related to Figs 4B and S5B are available under accession code https://doi.org/10.6084/m9.figshare.28781003. The Alphafold2 model of the Rs3367-CoV spike related to Figs 6 and S9 is available under accession code https://doi.org/10.6084/m9.figshare.30052111. Input parameters and model statistics for this Alphafold2 model related to Figs 6 and S9 are available at https://doi.org/10.6084/m9.figshare.30052138. GenBank accession numbers of sarbecovirus spike sequences with Y623H are available in S1 File. The Rs3367-CoV spike reference nucleotide sequence used to align the nanopore sequencing data is available in S2 File (plaintext file in FASTA format). Alignments from nanopore long-read sequencing of rVSV–Rs3367-CoV viral supernatants (binary files in BAM format) are available at the Sequence Read Archive (SRA; https://www.ncbi.nlm.nih.gov/sra/?term=PRJNA1313131) under Bioproject PRJNA1313131. Variant calls generated from these alignments are available in S3 File (Excel file format). The complete sequencing analysis workflow is available at https://github.com/chandranlab/tse2025.

**Funding:** This work was supported by the National Institutes of Health (NIH; https://www.niaid.nih.gov/) grants R01AI132633 and K08AI180364 to K.C. and E.H.M., respectively. E.H.M. was additionally supported by the Institute for Clinical and Translational Research at Einstein and Montefiore (https://einsteinmed.edu/centers/ictr) (K12TR004411). This work was also funded in part by a Welch Foundation grant (https://welch1.org/; F-0003-19620604) to J.S.M. A.L.T. and J.B. were additionally supported by the NIH training grants T32GM149364 (Medical Scientist Training Program) and T32AI070117 (Geographic

virus-receptor binding and entry could precede, or even initially replace, substitutions that enhance spike cleavage in the zoonotic host.

## Author summary

Recent epidemic-causing coronaviruses, including SARS-CoV-2, originated in bats. Large numbers of such viruses circulate in bats and pose clear and present risks to humans. However, our incomplete understanding of the variables that influence viral 'spillover' into new hosts challenges attempts to stratify viruses by threat level. We previously showed that the spike protein of the bat coronavirus SHC014-CoV, closely related to SARS-CoV-1 and SARS-CoV-2, primarily exists in a closed conformation that is incompatible with its binding to the viral receptor, ACE2, and that genetic changes in key control sequences 'open' the spike and unlock its entry activity. Here, we extend these findings to a second bat coronavirus. We demonstrate that WIV1-CoV, a highly studied virus that was previously isolated from the same population of Chinese horseshoe bats, likely acquired a genetic change in the same hotspot region of the spike during its propagation in cell culture. Our findings support the idea that the closed spikes of at least some (and likely, many) bat coronaviruses, while exquisitely adapted to their natural milieu in the bat digestive system, suffer poor functional activity outside this milieu, imposing a barrier to viral spillover that must be overcome through viral adaptation. Further, we identify spike genetic changes that overcome this deficit and may have value as prognostic markers of zoonotic risk.

## Introduction

Coronaviruses (CoVs) are enveloped, positive-strand RNA viruses that infect a wide range of hosts. In recent decades, three bat-origin CoVs—MERS-CoV, SARS-CoV, and SARS-CoV-2—have caused human epidemics, raising concerns about the spillover risks posed by the large number of related viruses known to circulate in bats [1,2]. Although CoV sequences have been detected and characterized in bat populations worldwide, only a small fraction of these viruses have been successfully isolated and/or investigated in detail, and the molecular factors that influence their zoonotic potential remain incompletely understood [1].

The coronavirus spike protein, S, forms membrane-embedded homotrimers displayed on the surface of viral particles. The precursor spike, S0, is post-translationally cleaved into two subunits, S1 and S2, which remain closely associated in the pre-fusion spike conformer. S1, the membrane-distal subunit, binds host cell receptors and regulates the conformational state of S2, the transmembrane subunit, which mediates viral membrane fusion [3–5]. The spike protein undergoes dynamic movements and conformational changes essential for viral entry into host cells. In one such movement, the receptor-binding domain (RBD) in S1 oscillates between "up"

Medicine and Emerging Infections), respectively, at Albert Einstein College of Medicine. The funders had no role in study design, data collection and analysis, decision to publish, or preparation of the manuscript. The content is solely the responsibility of the authors and does not necessarily represent the official views of the National Institutes of Health.

and "down" states. Transition from the sterically shielded "down" state to the exposed "up" state facilitates RBD-receptor binding, thereby initiating the process of viral entry. Conversely, the closed spike conformation (all RBDs down) shields key epitopes from neutralizing antibodies, whereas the open spike conformers (one or more RBDs up) are more sensitive to antibody neutralization [6]. The relative occupancies of these two states are influenced by multiple molecular and environmental factors that remain to be fully elucidated [6,7], and their modulation likely serves to optimize the tradeoffs between viral stability, infectivity, and immune evasion.

We and others have recently shown that, unlike the spikes of bat-origin CoVs associated with human epidemics, 'native' bat CoV spikes (e.g., BANAL-20–52, BANAL-20–236, RaTG13, and SHC014-CoV) predominantly exist in the closed conformation, with all three RBDs down [8–12]. Consistent with this observation, we showed that the WT SHC014-CoV spike binds only weakly to bat and human orthologs of its receptor, ACE2, despite its RBD's high intrinsic affinity for ACE2 [13]. Further, we identified novel amino acid substitutions in S1 and S2 sequences outside the RBD that enhanced the viral entry activity of the SHC014-CoV spike in cell culture by increasing its propensity to adopt an open conformation compatible with RBD-ACE2 recognition. We postulated that these amino acid changes represent genetic pathways that could allow bat CoVs to adapt to new environments in which increased RBD availability confers greater viral fitness [12].

An intriguing observation that we made during this previous work on the SHC014-CoV spike [12] was its divergent behavior from the spike of WIV1-CoV, a SARS-like bat CoV that closely resembles SHC014-CoV in amino acid sequence. WIV1-CoV was the first SARS-like CoV to be successfully isolated from a Chinese horseshoe bat, the putative reservoir host of SARS-CoV [14], and it could efficiently infect human airway cultures, raising concern that WIV1-CoV was 'poised for human emergence' [15]. WIV1-CoV circulates in the same colony of bats in Yunnan Province, China from which the SHC014-CoV genome was also discovered and sequenced; however, the latter virus was not recovered from these samples [14]. We found that, unlike the WT SHC014-CoV spike, the WT WIV1-CoV spike could readily bind human ACE2 and mediate viral entry, as observed previously [14–16] and consistent with recent structural observations that the WIV1-CoV spike has a more open conformation than its SHC014-CoV counterpart [12]. Herein, we show that a single amino acid residue difference in the S1 subunit can fully account for the divergent properties of the WIV1-CoV and SHC014-CoV spikes without altering their propensity to undergo proteolytic cleavage at the S1-S2 boundary. Strikingly, we found that WIV1-CoV differs from Rs3367-CoV—also sequenced from these samples [14]—at exactly the same amino acid position. Our analysis suggests that WIV1-CoV's more open, entry-competent spike is an artifact of viral adaptation to cell culture, and that its progenitor virus likely bears a spike resembling Rs3367-CoV's spike in amino acid sequence and function. We infer that WIV1-CoV's more open spike does not reflect its progenitor's authentic biology in bats—a more closed spike conformation may be optimal for the fitness of the WIV1-CoV progenitor (and other bat CoVs, such as SHC014-CoV) in their natural environments.

PLOS Pathogens

Moreover, we postulate that the acquisition of mutations that afford a more open spike conformation could precede adaptive changes that enhance spike cleavage and may possibly even render the latter unnecessary for zoonotic spill-over into non-bat mammals.

## Results

### A single amino acid substitution explains the difference in cell entry activity between the SHC014-CoV and WIV1-CoV spikes

We showed recently that the spike protein from SHC014-CoV is poorly competent for viral entry, and that its activity could be rescued by substitutions in the S1 N–terminal domain (NTD) and S2 fusion peptide-proximal region (FPPR) that enhance RBD-ACE2 engagement [12]. By contrast, the spike protein from a closely related sarbecovirus, WIV1-CoV, is highly entry-competent in its WT form [15]. This phenotypic difference was especially striking in light of the highly similar amino acid sequences of the SHC014-CoV and WIV1-CoV spikes (97% identical; S1A Fig) and the origin of these viruses in the same bat samples (see below). To investigate the molecular basis of their divergent behavior, we generated a panel of single-cycle vesicular stomatitis virus (scVSV) pseudotypes bearing spike chimeras in which SHC014-CoV NTD, RBD, and C-terminal domain 2 (CTD2) sequences were introduced into the WIV1-CoV spike (Figs 1A and S1). We measured

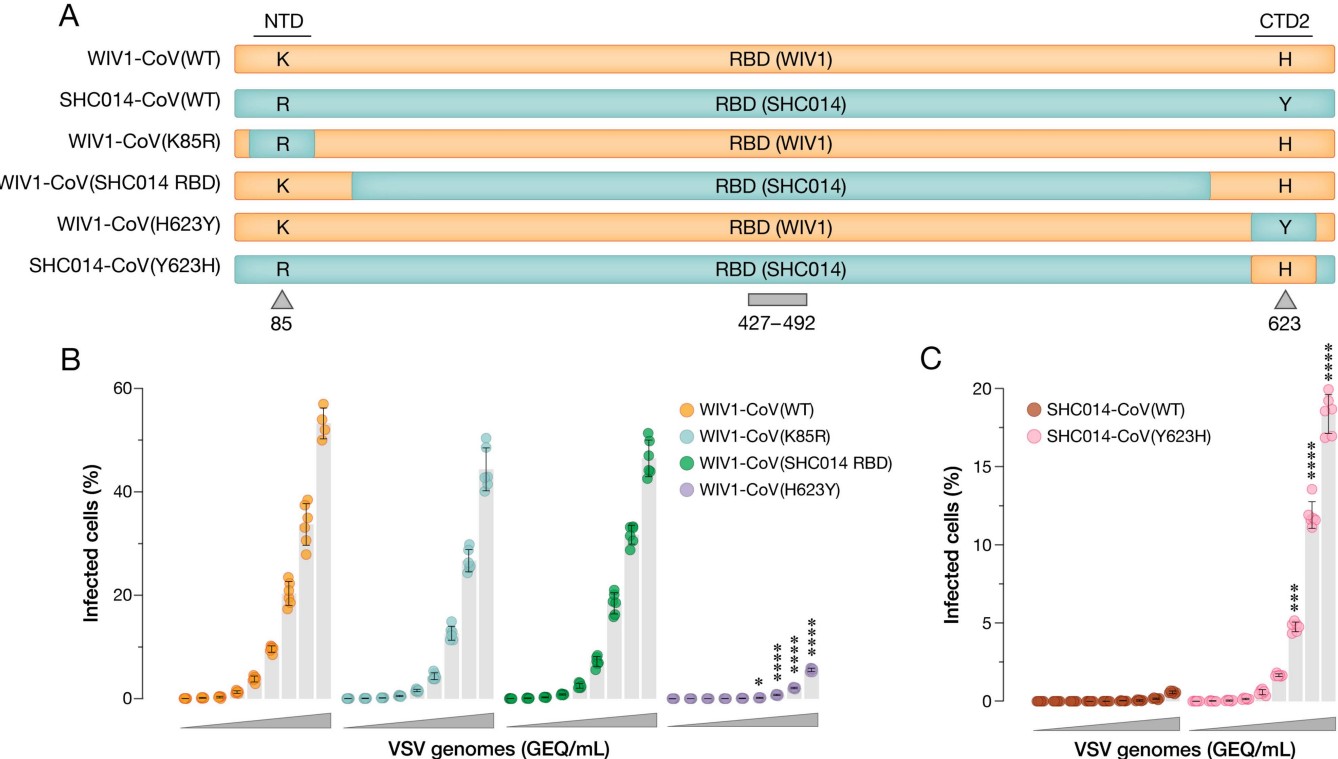

**Fig 1. Spike amino acid substitution Y623H causes binding and infectivity differences between WIV1-CoV and SHC014-CoV. (a)** Chimeric scVSV constructs in the spike NTD, RBD, or CTD2 regions are switched between WIV1-CoV and SHC014-CoV. **(b-c)** DBT-9 cells overexpressing *Ra*ACE2 were infected with genome-normalized amounts of scVSV-WIV1-CoV S (WT, K85R, SHC014 RBD, or H623Y) **(b)**, or scVSV-SHC014-CoV S (WT or Y623H) **(c)**. Infection was scored by eGFP expression at 16-18h post-infection (average±SD, n=6 from 3 independent experiments). A range of $3.98 \times 10^8$ to $6.06 \times 10^4$ viral genomes-equivalents (GEQ) was used. Groups were compared against WT with two-way ANOVA with Tukey's correction for multiple comparisons, ns p>0.05; ** p<0.01; *** p<0.001; **** p<0.0001.

the infectivity of this scVSV panel in ACE2-null DBT-9 murine astrocytoma cells expressing ACE2 from the intermediate horseshoe bat (*Rhinolophus affinis*; DBT-*Ra*ACE2) [17,18]. The CTD2 chimera, which replaced His with Tyr at position 623 (H623Y) in WIV1-CoV, greatly reduced infection, whereas the NTD and RBD chimeras had little effect (Fig 1B). Conversely, introduction of Y623H into the SHC014-CoV spike substantially boosted infection (Fig 1C). Therefore, a single amino acid residue in the spike CTD2 domain can fully explain the weak and strong cell entry activities of SHC014-CoV and WIV1-CoV, respectively, as also recently reported by Qiao and Wang [11].

## WIV1-CoV and Rs3367-CoV spike amino acid sequences differ at only two positions

We examined the original publication that described the isolation of WIV1-CoV [14] for possible clues to the genotype and behavior of this virus vis-à-vis that of SHC014-CoV. In that study, the investigators collected anal swabs and fecal samples from Chinese horseshoe bats in Yunnan Province, China and used them to determine the complete genome sequences of two novel SARS-like bat CoVs—RsSHC014-CoV (SHC014-CoV) and Rs3367-CoV. They then exposed supernatants from PCR-positive fecal samples to Vero E6 cells and successfully obtained a single viral isolate—WIV1-CoV—with 99.9% nucleotide sequence identity to the Rs3367-CoV genome sequence. We aligned the amino acid sequences of the Rs3367-CoV and WIV1-CoV spikes and found that the latter differs at only two positions: Y623H in the CTD2 domain of the S1 subunit and N1167D in the HR2 heptad repeat sequence of the S2 subunit (S2A Fig).

## Y623H was likely selected during WIV1-CoV isolation from bat fecal samples

We postulated that successful WIV1-CoV rescue required spike substitutions at one or both positions (Y623H and/or N1176D). Accordingly, we generated scVSVs bearing the spike protein of WIV1-CoV, Rs3367-CoV, Rs3367-CoV(Y623H) and Rs3367-CoV(N1167D) (Fig 2A). We first compared genome-normalized preparations of scVSVs bearing the spike proteins of Rs3367-CoV and WIV1-CoV in infectivity assays. scVSV–WIV1-CoV particles exhibited significantly greater infectivity than their Rs3367-CoV counterparts (Fig 2B), despite the similar levels of expression, cell-surface localization, and viral incorporation of these spikes (S1B–S1F and S8C Figs). Further, the Y623H substitution could enhance scVSV–Rs3367-CoV infection to scVSV–WIV1-CoV levels, whereas N1167D could not (Fig 2B). Sequence analysis of spike proteins from 5,944,141 sarbecovirus genomes revealed that Y623H is a highly nonconservative amino acid change that is essentially unique to WIV1-CoV (S3 Fig and S1 File), lending credence to the idea that it arose and/or was enriched under selective pressure.

A further implication of this hypothesis is that the generation of a replication-component VSV (rVSV) bearing the WT Rs3367-CoV spike as its only entry glycoprotein should not occur in this cell culture system, which closely resembles that used by Ge and co-workers [14]. Conversely, successful viral rescue should necessitate the acquisition of growth-adaptive mutation(s) in the spike gene. Accordingly, we performed two independent rescue experiments using the well-established VSV plasmid-based reverse genetics system and successfully obtained replicating viral populations in each attempt. Long-read sequencing of viral RNA-derived reverse transcription-PCR (RT-PCR) products uncovered four highly enriched missense mutations (encoding V584I and V946G in population 1, and Q836P and N943H in population 2) (S3 File). Remarkably, these novel substitutions mapped to two sites previously shown to influence spike conformational dynamics and RBD availability—the first in, or structurally apposed to, the S2 fusion peptide-proximal region (FPPR; Q836P, N943H, V946G), and the second (V584I) in the S1 CTD2 subdomain adjacent to the Y623H-containing '630 loop' (see Fig 6 and Discussion). Indeed, we recently showed that a spike-opening substitution, A835D, in the spike FPPR afforded rVSV–SHC014-CoV rescue [12]. These findings, together with the capacity of the rVSV–WIV1-CoV to replicate in cell culture without the acquisition of any spike substitutions at a high frequency [12], provide additional evidence that the Rs3367-CoV spike is greatly impaired for viral entry in cell culture in its WT form. We conclude that Y623H is a genetic adaptation selected during outgrowth of a Rs3367-CoV–like virus from bat fecal samples and that WIV1-CoV is a cell culture-adapted variant of this progenitor virus.

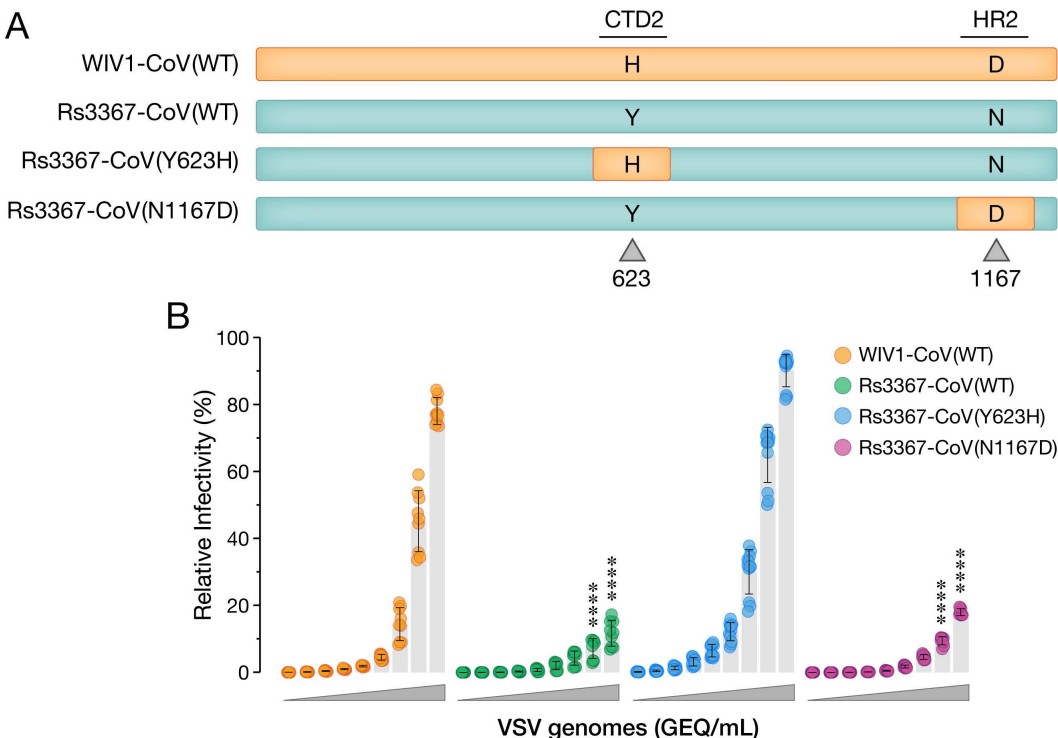

**Fig 2. Spike amino acid substitution Y623H drives receptor-dependent differences in viral entry between scVSV-Rs3367-CoV and scVSV-WIV1-CoV. (a)** Schematic of generated chimeric scVSV constructs in which single amino acid substitutions at positions 623 and 1167 were exchanged between WIV1-CoV and Rs3367-CoV spikes. **(b)** DBT-9 cells overexpressing *Ra*ACE2 were infected with genome-normalized amounts of scVSV-WIV1-CoV, scVSV-Rs3367-CoV, scVSV-Rs3367-CoV(Y623H), or scVSV-Rs3367-CoV (N1167D). Infection was scored by eGFP expression at 16–18 h post-infection (average±SD, n=6-12 from 2-4 independent experiments). A range of $2.19 \times 10^{10}$ to $3.33 \times 10^{6}$ viral genome-equivalents (GEQ) was used. Groups were compared against WT with two-way ANOVA with Tukey's correction for multiple comparisons, ns p>0.05; ** p<0.01; *** p<0.001; **** p<0.0001.

### Y623H enhances Rs3367-CoV spike:ACE2 recognition

These findings raised the possibility that the Rs3367-CoV spike, like SHC014-CoV but not WIV1-CoV, predominantly assumes a closed "all RBDs down" conformation that cannot bind its cellular receptor, ACE2, and that Y623H rescues viral entry by modulating spike dynamics and RBD availability. To investigate this, we coated ELISA plates with soluble human ACE2 (*Hs*ACE2) and captured genome-normalized equivalents of scVSV particles bearing spikes from Rs3367-CoV, WIV1-CoV, and our chimeric constructs. Viral capture was then detected using a spike-binding monoclonal antibody (mAb). Commensurate with their infectivity in DBT-*Ra*ACE2 (Fig 2) and DBT-*Hs*ACE2 cells (see S10 Fig), the Rs3367-CoV and WIV1-CoV spikes exhibited low and high levels of dose-dependent binding to *Hs*ACE2, respectively. The Tyr⇓◊His substitution at residue 623 could fully recapitulate the differences in spike:ACE2 recognition not only between Rs3367-CoV and WIV1-CoV, but also between SHC014-CoV and WIV1-CoV (Figs 3A and S2B–S2C). We conclude that Y623H enhances WIV1-CoV fitness by increasing its capacity to bind ACE2 during cell entry.

### Y623H increases Rs3367-CoV RBD availability for ACE2 binding

Because the RBDs of Rs3367-CoV and WIV1-CoV do not differ in amino acid sequence and the latter has been shown to recognize *Hs*ACE2 with nanomolar affinity [13], we reasoned that Y623H likely enhances spike:ACE2 binding by increasing the propensity of the WIV1-CoV spike to sample the RBD-up conformation ('RBD availability'), as we also recently observed for SHC014-CoV spikes bearing growth-adaptive mutations [12]. To test this, we first compared the sensitivities

of scVSVs bearing Rs3367-CoV, WIV1-CoV, and their chimeras to neutralization by Adagio-2 (ADG-2), a monoclonal antibody (mAb) that specifically recognizes the "up" conformer of the RBD [19,20]. Consistent with our hypothesis, ADG-2 neutralized scVSV–WIV1-CoV particles much more potently than scVSV–Rs3367-CoV (Fig 3B). Moreover, ADG-2's activity against the chimeras reflected the spike genotype at residue 623 and not 1167, as seen in the infectivity and ACE2-binding studies above. Residue 623 similarly influenced SHC014-CoV spike sensitivity to ADG-2 neutralization (S2D–S2E Fig), indicating that its biochemical and functional effects are transferable to the SHC014-CoV genetic background.

Importantly, we could directly corroborate these inferred effects of Y623H on spike RBD positioning and availability in a flow-cytometric assay. Specifically, both ADG-2 (Figs 3D, S4A, and S8A) and *Hs*ACE2 (Figs 3C, S4B, and S8B) exhibited increases in binding to cells displaying the WIV1-CoV spike relative to those displaying the Rs3367-CoV spike. By contrast, mAb S309, whose epitope is available in both "up" and "down" RBD conformers [21], exhibited similar levels of binding to both cell populations (S1B–S1C and S8C Figs), confirming that the Rs3367-CoV and WIV1-CoV spikes were expressed at similar levels at the cell surface.

Together, these findings strongly support our conjectures that: (i) the Y623H substitution was selected during cell culture adaptation of a progenitor virus with an Rs3367-CoV–like spike culture, leading to WIV1-CoV; and (ii) Y623H promotes viral fitness by increasing the RBD availability of the WIV1-CoV spike, thereby enhancing its capacity to recognize ACE2 during cell entry.

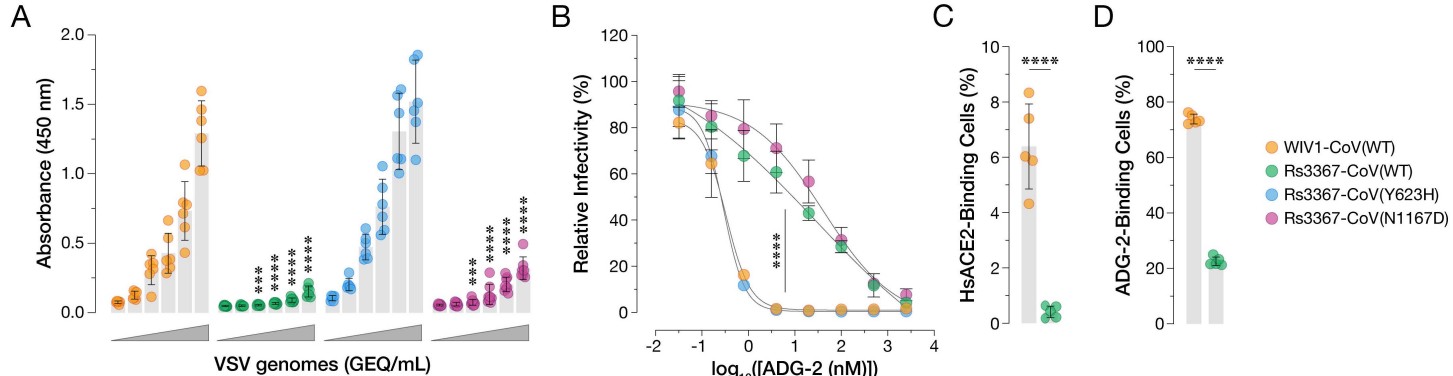

**Fig 3. Y623H increases *Hs*ACE2 binding and RBD availability by scVSV-Rs3367-CoV. (a)** Genome-normalized amounts of scVSV particles bearing spikes of WIV1-CoV, Rs3367-CoV, Rs3367-CoV(Y623H), or Rs3367-CoV(N1167D) were diluted with 3-fold dilutions onto ELISA plates precoated with soluble *Hs*ACE2, followed by a spike-specific mAb, and incubation with an anti-human HRP-conjugated secondary antibody (average±SD, n = 6-8 from 3-4 independent experiments). A range of $2.4 \times 10^7$ to $9.9 \times 10^4$ viral GEQ was used. Groups were compared against WT with two-way ANOVA with Tukey's correction for multiple comparisons, ns $p > 0.05$; ** $p < 0.01$; *** $p < 0.001$; **** $p < 0.0001$. **(b)** Pre-titrated amounts of scVSV-WIV1-CoV, scVSV-Rs3367-CoV, scVSV-Rs3367-CoV(Y623H), or scVSV-Rs3367-CoV(N1167D) were incubated with serial 3-fold dilutions of ADG-2 mAb, starting at 2.5 µM, for 1 h at 37˚C. Virus:ADG-2 mixtures were applied to monolayers of DBT-9 cells overexpressing *Rs*ACE2. At 16-18 hours post-infection, cells were fixed, and infected cells were scored by eGFP expression (average±SD, n = 9 from 3 independent experiments). Relative infectivity (%) was calculated by normalizing to a no-mAb control for each virus. AUC values were calculated for each curve, and groups were compared by one-way ANOVA with Dunnett's post hoc test, ns $p > 0.05$; ** $p < 0.01$; *** $p < 0.001$; **** $p < 0.0001$. Only the statistically significant comparisons are shown. **(c-d)** 293T cells were transfected with plasmids expressing either WIV1-CoV or Rs3367-CoV spikes and harvested 24 h post-transfection. **(c)** Cells were incubated with 50 nM of soluble *Hs*ACE2 and bound protein was detected with Strep-Tactin XT PE. Binding was assessed by flow cytometry (average±SD, n = 5 from 3 independent experiments). **(d)** Cells were immunostained for cell surface expression by ADG-2, followed by a fluorescent secondary antibody, and analyzed using flow cytometry (average±SD, n = 5 from 3 independent experiments). WIV1-CoV vs. Rs3367-CoV were compared with unpaired t-test with Welch's correction, ns $p > 0.05$; ** $p < 0.01$; *** $p < 0.001$; **** $p < 0.0001$.

**Rs3367-CoV and WIV1-CoV spikes are not cleaved in producer cells but undergo cleavage by exogenous trypsin treatment**

Post-translational cleavage of the spike precursor S0 into S1 and S2 'primes' the spike for cell entry by enhancing spike dynamics and liberating the N-terminal fusion peptide in S2, which inserts into the target cellular membrane during viral membrane fusion [9,22,23]. Previous work has shown that the spike proteins of most bat coronaviruses lack a polybasic cleavage site for proprotein convertases (e.g., furin) at the S1–S2 junction and are thus not cleaved in virus-producer cells [8,24]. The Rs3367-CoV spike also lacks a consensus proprotein cleavage site, indicating that its lack of cleavage may account for the inability of its RBDs to readily adopt the ACE2-accessible "up" conformation during viral entry. To test this hypothesis, we first transfected 293T cells with expression plasmids encoding either the WT or mutant Rs3367-CoV spikes. Cells were then harvested, lysed, and analyzed by western blot using an S2-directed antibody. Only a band corresponding to the uncleaved S0 precursor was observed for all these proteins (Fig 4A). Similar results were observed with the spike proteins from WIV1-CoV (Fig 4A) and SHC014-CoV (S5A Fig), as reported previously. By contrast, the SARS-CoV-2 spike was cleaved in the transfected cells, as expected [25,26].

Many bat CoV spikes, including those of Rs3367-CoV, WIV1-CoV, and SHC014-CoV, harbor a tryptic cleavage site at the S1-S2 junction, and an extensive body of evidence indicates that exogenous trypsin treatment can effect this priming step [24,27,28]. Accordingly, we incubated spike-expressing cells with increasing amounts of trypsin for 1h at 37°C and evaluated spike protein cleavage by western blot as above. Both Rs3367-CoV and WIV1-CoV spikes were cleaved by trypsin, with no apparent differences in the kinetics or extent of cleavage (Fig 4B). Similar results were observed with WT and Y623H SHC014-CoV spikes (S5B Fig). We conclude that the Y623H substitution does not grossly alter the propensity of these bat CoV spikes to undergo S1/S2 cleavage by trypsin.

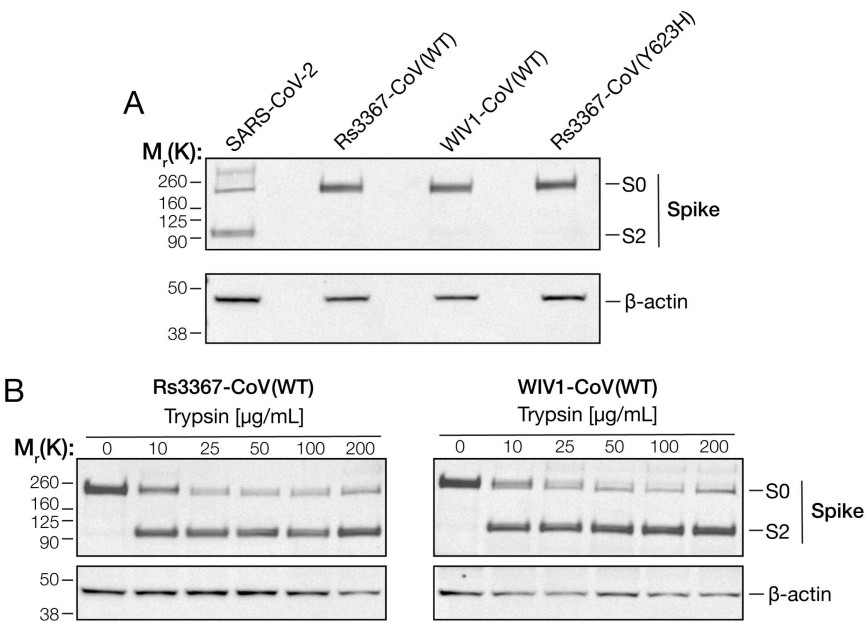

**Fig 4. Trypsin treatment cleaves bat CoV spikes at the S1–S2 boundary. (a-b)** 293T cells were transfected with plasmids to express the indicated S proteins. 24 hours post-transfection, cells were harvested, lysed, and S protein expression was analyzed by immunoblot with a CoV S2-directed polyclonal antibody. Bands corresponding to uncleaved spike (S0) or the cleaved S2 subunit (S2) are indicated based on their relative molecular weights ($M_r$). β-actin was included as a loading control. **(b)** At 24 h post-transfection, spike-expressing cells were incubated with increasing amounts of trypsin for 1 h at 37°C.

## Trypsin cleavage-mediated boost in cell entry by the Rs3367-CoV spike is associated with enhanced RBD availability

The S1/S2 cleavage can prime multiple steps in coronavirus entry by virtue of its complex structural and functional effects on the spike [9,22,26,29,30]. Concordantly, previous work has shown that exogenous trypsin treatment of bat CoV spike proteins (and attendant S1/S2 cleavage) enhances cell entry [31–33]. To test if S1/S2 cleavage of the Rs3367-CoV, WIV1-CoV, and SHC014-CoV spikes affects entry, scVSVs bearing these spikes were exposed to trypsin and spun onto Vero cells at 4°C for 1h, as described previously [32]. We observed substantial trypsin-dependent increases in the entry activities of all three spikes (Figs 5A and S6).

We next used our cell-based spike-display assay to specifically examine if trypsin cleavage modulates spike:ACE2 recognition. Spike-transfected cells were exposed to trypsin for 1h at 4°C and then incubated with soluble *Hs*ACE2 prior to analysis by flow cytometry, as above. Both Rs3367-CoV and WIV1-CoV spikes exhibited increased *Hs*ACE2 binding after trypsin treatment (Figs 5B and S7A–S7B). Strikingly, *Hs*ACE2 binding to Rs3367-CoV approached levels similar to those seen with WIV1-CoV in the absence of trypsin. We conclude that S1/S2 cleavage boosts viral entry at least in part by increasing the propensity of the Rs3367-CoV spike to sample the 'RBD-up' conformation, thereby enhancing its availability to bind to ACE2. Moreover, we infer that the Y623H substitution was acquired and/or selected during outgrowth of a Rs3367-CoV–like virus from bat fecal samples as an adaptation to increase RBD availability for ACE2 recognition in the absence of proteolytic priming (also see Discussion).

## Discussion

The ongoing discovery of beta-CoVs in bats around the world with high sequence similarity to outbreak- and pandemic-causing CoVs has raised concerns over the potential for new spillover events [34–38]. The spike protein is a major determinant of the zoonotic potential of coronaviruses. A large body of evidence indicates that molecular

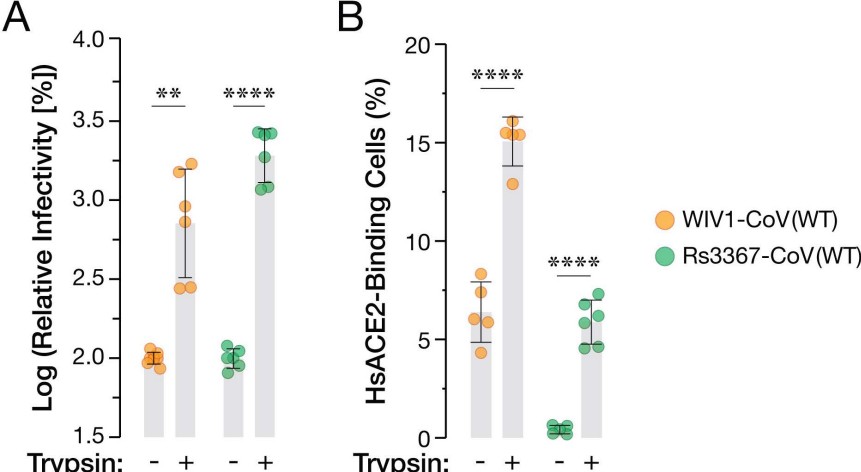

**Fig 5. Trypsin treatment increases ACE2-dependent cell entry and ACE2 binding by bat CoV spikes. (a)** scVSVs bearing spikes of WIV1-CoV or Rs3367-CoV were mixed with 200 µg/mL of trypsin and added onto Vero cells. At 16-18 hours post-infection, infection levels were scored by eGFP expression (average±SD. n = 6 from 3 independent experiments). Infectivity values were normalized to no-trypsin infection controls. **(b)** 293T cells were transfected with plasmid expression vectors encoding either WIV1-CoV or Rs3367-CoV spike proteins. Spike-expressing cells were incubated with trypsin at 5 µg/mL for 1h at 4°C. Spike binding was assessed by mixing with soluble *Hs*ACE2 for 1h at 4°C, followed by incubation with Streptactin PE and analysis by flow cytometry. (average±SD. n = 5-6 from 3 independent experiments). Groups (no trypsin vs. trypsin treatment) were compared with unpaired t-test with Welch's correction, ns p > 0.05; ** p < 0.01; *** p < 0.001; **** p < 0.0001.

incompatibilities at the spike RBD-receptor interface pose barriers to interspecies transmission [6,39]. Further, studies on the trajectory of SARS-CoV-2 evolution have identified suites of genetic adaptations that modulate spike stability, spike opening/RBD availability, proteolytic cleavage, membrane fusion, and vulnerability to antibody blockade in an interwoven manner [29,40]. These adaptations were presumably selected in response to the changing selective landscape encountered by the virus over the course of the COVID-19 pandemic. Specifically, earlier SARS-CoV-2 lineages that circulated largely in immunologically naïve populations are characterized by relatively open spikes that are highly competent to engage their receptor, ACE2, and mediate cell entry, whereas later lineages (e.g., Omicron) that encountered much greater immune pressure in their SARS-CoV-2–experienced hosts predominantly possess closed spikes [41,42]. These findings imply that the bat sarbecovirus ancestral to SARS-CoV-2 likely bears a spike that adopts a relatively closed conformation to minimize its susceptibility to antibody neutralization in its natural host. They also support a model in which SARS-CoV-2's successful spillover into humans (or an intermediate host) was initially driven in part by genetic adaptations, especially the acquisition of a furin cleavage site at the S1-S2 boundary, which enhanced spike opening, RBD:receptor recognition, and viral entry [43].

Recent work has corroborated and extended this idea by showing that the spikes of beta-CoVs circulating in bats are not cleaved in virus/spike-producer cells and largely adopt closed conformations [8,9]. Perhaps more surprisingly, we observed that the spike from at least one such virus—SHC014-CoV— transitioned so rarely from a closed to an open state that it was essentially unable to bind ACE2 and mediate viral entry in cell culture. Using a recombinant VSV-based forward-genetic approach, we previously identified two novel spike-opening substitutions in S1 and S2 sequences outside the RBD (in the NTD and FPPR, respectively) that enhanced RBD-ACE2 binding and viral entry, but at the expense of dramatic increases in antibody susceptibility [12]. The behavior of SHC014-CoV stood in stark contrast to the high entry activity of the spike from the closely related WIV1-CoV, which was isolated from the same colony of horseshoe bats in Yunnan Province, China [14]. Here we found that, as expected, the WT WIV1-CoV spike phenocopies the VSV-adapted SHC014-CoV spike in its enhanced capacity to recognize ACE2 and its increased susceptibility to antibody neutralization. Concordantly, recent structural work indicates that the WIV1-CoV spike can adopt the partially open 'one-RBD-up' state, whereas the SHC014-CoV spike cannot [11,12]. These findings pointed to one or more amino acid sequence differences between SHC014-CoV and WIV1-CoV as responsible for the correlated phenotypic differences between the spikes of these rhinolophid bat beta-CoVs.

We initially postulated that the SHC014-CoV:WIV1-CoV difference would map to one of the sequences (FPPR, NTD) associated with SHC014-CoV spike adaptation. Instead, our analysis of inter-spike chimeras identified a single determinative amino acid difference—Y623H—in the C-terminal domain 2 (CTD2) of the S1 subunit, a finding also reported by Qiao and Wang [11] while this manuscript was in preparation. In seeking to uncover the origin of this amino acid change, we examined the original publication that described the isolation of WIV1-CoV [14]. From our analysis of the Rs3367-CoV spike, we infer that the Y623H change, very rarely observed among sarbecoviruses, was selected during viral outgrowth of an Rs3367-CoV–like predecessor from the bat samples precisely because it enhanced infectivity via a set of biochemical changes similar to those we observed previously in the VSV-adapted SHC014-CoV spike—increases in spike opening, RBD availability, ACE2 engagement, and cell entry. Thus, the behavior of the spikes from Rs3367-CoV (and WT SHC014-CoV) likely reflects the properties of the authentic viruses in their natural hosts, whereas the behavior of the WIV1-CoV spike likely does not.

Residue 623 is located in the '630 loop,' a flexible sequence in CTD2 (Fig 6A and 6B). Positioned beneath CTD1 and the RBD, and packing between the NTDs of two neighboring protomers, the CTD2 plays a key role in regulating spike conformation [44]. Extensive work with the SARS-CoV-2 spike indicates that multiple steps in viral entry, including spike opening, cleavage, and receptor-binding–associated conformational changes culminating in membrane fusion are regulated by a bidirectional allosteric axis comprising CTD2, the C-terminal domain 1 (CTD1), and RBD in S1; and the FPPR in S2 [3,7]. Especially germane to this discussion, an order-to-disorder transition in the 630 loop correlated with increased

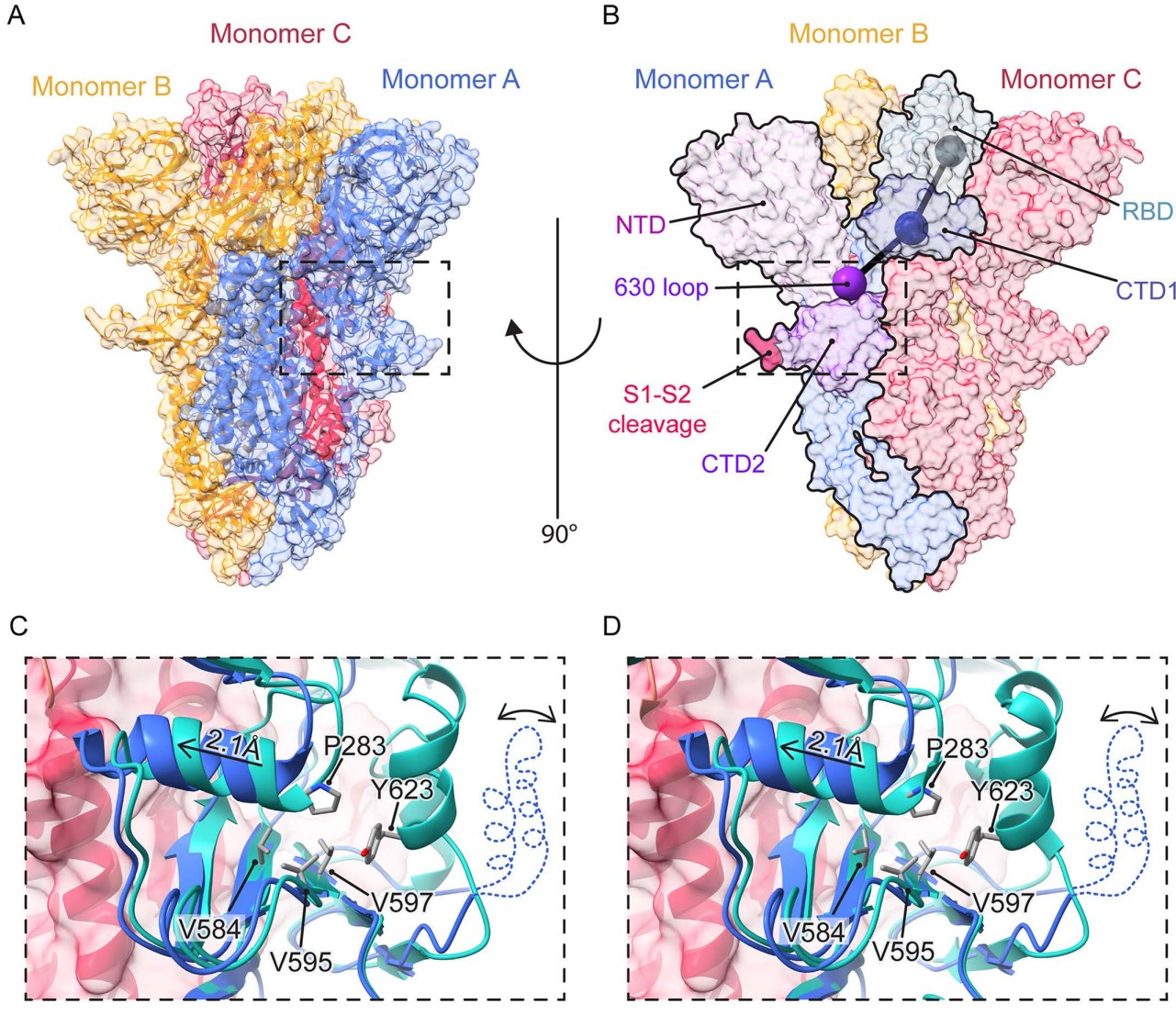

**Fig 6. Spike amino acid substitution Y623H triggers a conformational change leading to opening of the RBDs. (a-b)** Overview of the WIV1-CoV spike trimer (PDB: 8WQ0 [11]). The boxed region corresponds to the 630 loop in monomer **A. (b)** Spheres correspond to centroids of domains connecting the 630 loop (purple) to the RBD (grey) through the CTD1 (dark blue). **(c-d)** Close-up view showing the 630 loop region. **(c)** Structural superposition of WIV1-CoV (blue, PDB: 8WQ0) and Rs3367-CoV (cyan, AlphaFold2 model). **(d)** Structural superposition of WIV1-CoV (blue) and SHC014-CoV (cyan, PDB: 8WLU [11]). The missing 630 loop in WIV1-CoV is represented with a dotted line.

spike dynamics and a less stable, 'hair-trigger' spike protein with a greater propensity to undergo fusion-related rearrangements (and vice-versa). The substitution D614G in the SARS-CoV-2 spike, located just N-terminal to the 630 loop, stabilizes the furin-cleaved spike against premature inactivation by promoting the ordering of the 630 loop, thereby enhancing viral replication and transmissibility [45–48]. We infer based on available structures (also see below) that the Y623H substitution in Rs3367-CoV (and SHC014-CoV) does just the opposite—mobilizes the 630 loop in CTD2 to promote spike dynamics that drive RBD availability, receptor binding, and S2 conformational changes.

To further investigate the molecular basis of Y623H's effects on the bat CoV spikes, we used a structure of the SHC014-CoV spike to generate an AlphaFold2 (AF2) model of the Rs3367-CoV spike. We then superimposed the

structure of SHC014-CoV spike and the AF2 model of the Rs3367-CoV spike onto a previously solved structure of the WIV1-CoV spike [11] (Fig 6C and 6D). Consistent with the model discussed above, the 630 loop was disordered in WIV1-CoV but ordered in the Rs3367-CoV AF2 spike model (Fig 6C), as shown by us and others for the SHC014-CoV spike [11,12] (Fig 6D). As described previously for SARS-CoV-2 S(D614G), the ordered 630 loop in these bat spikes packs along a largely solvent-exposed hydrophobic surface comprising the upper β-sheet of the CTD2 (Val595 and Val597) and Pro283 (Pro295 in SARS-CoV-2) from the NTD-RBD linker [46]. In SARS-CoV-2, the residue corresponding to Tyr623 (Tyr636) appears key to this interaction. Accordingly, we propose that Y623H disrupts the interaction of the 630 loop with the CTD2 β-sheet and the NTD-RBD linker, increasing its mobility, as evidenced by a noticeable shift of the NTD-RBD linker towards the core of the trimer (Fig 6C and 6D). Strikingly, our rVSV–Rs3367-CoV rescue experiments uncovered a substitution in a different residue—Val584 (to Ile)—adjacent to Val595 and Val597 in the CTD2 upper β-sheet. This V584I change would also be expected to destabilize the CTD2:630 loop:NTD-RBD linker interaction. We propose that these changes propagate upward to CTD1 and laterally to the FPPR, as described by Cai and Calvaresi [3,7], enhancing RBD availability and priming S2 to undergo its fusion-related conformational changes. Interestingly, the amino acid substitutions in the NTD-RBD linker and the FPPR that we previously identified in the VSV-adapted SHC014-CoV spike [12] and the FPPR substitutions we identified in the VSV-adapted Rs3367-CoV spike herein (S9 Fig) bracket the CTD2 and 630 loop substitutions (V584I and Y623H, respectively), providing further support for the SARS-CoV-2 allosteric model [3, 7].

It is well established that proteolytic cleavage of coronavirus spikes at the S1–S2 sequence boundary primes them to undergo membrane fusion-related conformational changes, thereby driving viral entry. More unexpectedly, our work herein and by others [10,11] indicates that the spikes of some bat-origin CoVs are 'locked down' in their uncleaved form and that they cannot engage their receptor to bind cells and initiate entry. This receptor-binding defect arises directly from the lack of S1–S2 cleavage, as shown by our observation that trypsin treatment (and attendant spike cleavage) enhances spike-ACE2 recognition in a virus-free experimental system. We surmise that the behavior of the SHC014-CoV and Rs3367-CoV spikes reflects not only their imperatives to maintain a high degree of biochemical stability and resistance to circulating antibodies in their native environment, the intestinal tract of insectivorous bats, but also their adaptation to a key feature of this environment—the high levels of extracellular trypsin-like serine proteases. Our findings raise the possibility that these adaptations are broader properties of bat sarbecoviruses than currently recognized.

This study adds to a growing body of work showing that the availability of host proteases to mediate the spike priming and/or triggering cleavages likely imposes barriers to the expansion of coronavirus tissue tropism and host range [49–51]. One such barrier may arise from the relatively poor activity of the cell-surface transmembrane protease TMPRSS2—an important player in the proteolytic priming of divergent viruses in the mammalian respiratory tract—against the monobasic S1-S2 cleavage sites of bat sarbecoviruses [25]. Although studies in more physiologically relevant model systems are needed, our findings suggest that spike-opening substitutions like Y623H (and the FPPR and NTD mutations we identified previously [12]), which promote receptor binding by uncleaved spikes, could overcome a barrier to extracellular spike cleavage by driving viral internalization into the endocytic pathway [52,53]. Here, endo/lysosomal proteases such as cathepsin L could cleave at both S1-S2 and S2′ sites to trigger viral membrane fusion [23,54–57]. This idea is consistent with the known capacity of diverse coronaviruses, including SARS-CoV, SARS-CoV-2, and mouse hepatitis virus, to use cysteine cathepsins for endosomal entry [54,56,58]. Both scVSV-Rs3367 WT and the Y623H mutant remained sensitive to $NH_4Cl$, a lysosomotropic compound that inhibits acid-dependent endosomal proteolytic activities, suggesting that both spikes predominantly use the endosomal pathway for entry in this cell culture system (S10 Fig).

The current paradigm for SARS-CoV-2 emergence posits that its bat precursor overcame a host protease barrier to viral entry in the mammalian respiratory tract by acquiring a polybasic furin cleavage site that also destabilized the spike, necessitating initial counter-adaptations such as D614G to enhance spike stability. Further adaptations then served to mask the spike against host neutralizing antibodies (including by closing the spike, as observed in Omicron variants) [41,42]. Our findings suggest the possibility of alternative spillover scenarios in which spike-opening substitutions that

promote virus-receptor binding and entry precede, or even initially replace, substitutions that enhance spike cleavage in the zoonotic host.

## Materials and methods

### Cell lines

Human embryonic kidney 293T cells (Thermo Fisher) were cultured in high-glucose Dulbecco's Modified Eagle Medium (DMEM, Thermo Fisher) supplemented with 10% heat-inactivated fetal bovine serum (FBS, Gemini), 1% GlutaMAX (Thermo Fisher), and 1% penicillin-streptomycin (P/S, Thermo Fisher). African green monkey kidney Vero Cells (American Type Culture Collection (ATCC)) were cultured in high-glucose DMEM supplemented with 2% FBS, 1% GlutaMAX, and 1% P/S. DBT-9 cells (gift of Ralph Baric, source unknown) were cultured in Minimum Essential Medium α (MEMα, Thermo Fisher) supplemented with 10% FBS, 1% GlutaMAX, 1% P/S, and 1% Amphotericin (Thermo Fisher). African green monkey kidney Vero81 cells knocked out for ACE2 stably transfected to express empty vector or *Rs*ACE2 (gift of Ralph Baric, source unknown) were cultured in high-glucose DMEM supplemented with 5% FBS, 1% non-essential amino acids (NEAA, Thermo Fisher), and 1% P/S. Cells overexpressing ACE2 were also cultured with 5 μg/mL puromycin (Gibco). All cell lines were grown in a humidified 37˚C incubator supplied with 5% $CO_2$ and subcultured every 2–3 days using 0.05% Trypsin/EDTA (Gibco).

### Plasmids

Plasmids encoding human codon-optimized spikes of SHC014-CoV (GenBank accession number KC881005.1), WIV1-CoV (GenBank accession number KC881007), and Rs3367-CoV (GenBank accession number KC881006.1), as well as relevant mutant and chimeric constructs, were generated and cloned into pTwist mammalian expression vectors.

### Generation of pseudotyped VSVs

Single cycle VSV (scVSV) pseudotypes bearing an eGFP reporter and spike proteins of SHC014-CoV, Rs3367-CoV, WIV1-CoV, or chimeric constructs of each, were produced in 293T cells as previously described [59–61]. Briefly, 293T cells were transfected with an expression plasmid encoding respective S proteins. Forty-eight hours later, cells were infected with a passage stock of VSVG/ΔG for 1 h at 37°C. Cells were subsequently washed eight times with high glucose DMEM to remove residual VSV-G. Viral supernatant was harvested forty-eight hours later and pelleted by a two-step ultracentrifugation.

### Generation of recombinant VSVs

A plasmid encoding the VSV genome was modified to replace its glycoprotein, G, with a human codon-optimized spike glycoprotein gene of Rs3367-CoV encoding a 21-amino acid C-tail truncation (S2 File). The VSV genome also encodes for an eGFP reporter gene as a separate transcriptional unit. rVSV–Rs3367-CoVs were generated via a plasmid-based rescue system in 293FT cells as described previously [12,62]. Briefly, 293FT cells were transfected with the VSV plasmid and plasmids expressing T7 polymerase and VSV N, P, M, G, and L proteins using polyethylenimine. At 48 h post-transfection, supernatants from the transfected cells were transferred to Vero cells. All rescues were conducted at 37˚C and no exogenous trypsin was added to the cell cultures. Viral growth was monitored by an eGFP reporter every day. Spike sequences were amplified from viral genomic RNA by RT-PCR and analyzed by nanopore sequencing (see below).

### Nanopore sequencing and analysis

Viral RNA was isolated from rescue population supernatants (Zymogen Quick-RNA viral Kit). cDNA was then generated through reverse transcription with a VSV genome-specific primer (forward 5'-CTCCAGCGGTATTGGCAGAT

-3', reverse 5'CCACATCGAGGGAATCGGAA-3') (Invitrogen Superscript IV). cDNA was amplified by PCR with VSV-specific primers flanking the spike gene (forward: 5'-AGGCCTTAATGTTTGGCCTG-3'; reverse: 5'AAATCATTGAACTCGTCGGTCTC-3') and NEB Q5 Hotstart master mix. Following this, long-read DNA sequencing on the Oxford Nanopore Technologies (ONT) platform (Plasmidsaurus) was performed to identify mutations present in the viral population.

Nanopore sequencing reads were aligned to a reference sequence (codon-optiimized Rs3367-CoV spike; S2 File) using minimap2 (https://github.com/lh3/minimap2) [63]. Genotype likelihoods and variant calls in the resulting alignment files (.bam) were determined using bcftools (https://www.htslib.org/) [64], and the sequence alignments and variant calls (S3 File) were visualized with Integrative Genomics Browser (IGV; https://igv.org/) [65]. The complete analysis pipeline is available at https://github.com/chandranlab/tse2025. The sequencing alignments are available in the Sequence Read Archive (SRA) under Bioproject PRJNA1313131.

### *Hs*ACE2 expression and purification

Amino acids 1–615 of HsACE2 were coupled to a PreScission site, 8X histidine tag, and Strep Tag II and cloned into a pαH vector. HsACE2 was expressed by polyethylenimine-induced transient transfection of FreeStyle 293-F cells (Thermo Fisher). After 6 days, the cell supernatant was harvested by centrifugation and clarified via passage through a 0.22 μm filter. *Hs*ACE2 was then purified from filtered supernatant via gravity flow over StrepTactin XT resin (IBA) followed by gel-filtration chromatography on a Superdex 200 Increase 10/300 column (GE Healthcare) into a buffer consisting of 2 mM Tris pH 8.0, 200 mM NaCl, 0.02% NaN3.

### Detection of CoV spike incorporation by ELISA

High-protein binding half-area 96-well plates (Corning) were coated with 3-fold serial dilutions of scVSV-CoVs starting at $1 \times 10^7$ genome copies/well overnight at 4°C. Plates were washed with PBS and blocked with 5% milk in PBS for 1h at 37°C. 50nM of mAb S309 was incubated for 1 hour at 37°C. Plates were washed and goat anti-human IgG secondary antibody conjugated to horseradish peroxidase (HRP) was added and incubated for 1h at 37°C. Plates were washed, then 1-Step Ultra TMB-ELISA Substrate Solution (Thermo Fisher) was added and quenched with 0.5 M $H_2SO_4$. Absorbance was read at 450 nm.

### Detection of CoV spike surface expression by flow cytometry

293T cells were seeded in a 10-cm dish. 20 h later, cells were transfected with 5 μg of a plasmid encoding an expression vector and S protein of either WIV1-CoV, Rs3367-CoV, SHC014-CoV, or SHC014-CoV Y623H. At 20 h post-transfection, cells were harvested and stained with a spike-specific mAb—either S309 (20 μg/mL) or ADG-2 (20 μg/mL)—followed by anti-human Alexa Fluor 488 (4 μg/mL) for 1 h at 4°C. After washing, stained cells were filtered through a 41-μm nylon net filter (Millipore) and analyzed using a Cytek Aurora Flow Cytometer and FlowJo software.

### Detection of CoV spike surface expression and cleavage status by western blot

293T cells were seeded in six-well plates. 24 h later, cells were transfected with 2 μg of expression plasmids encoding SARS-CoV-2, WIV1-CoV, Rs3367-CoV (WT or Y623H), SHC014-CoV (WT or Y623H). For cells exposed to trypsin, at 24 h post-transfection, cells were incubated with increasing concentrations of trypsin (Gibco, Cat# 15050057) for 1 h at 37°C. Trypsin was stopped by adding 10% FBS for 15 min at 37°C. Cells were harvested and lysed with RIPA Lysis and Extraction Buffer (Thermo Scientific Cat# 89900). Cell lysates were analyzed by western Blot using Bolt 12% Bis-Tris Plus protein gels (Invitrogen) followed by protein transfer to nitrocellulose membranes using the iBlot2 Gel Transfer Device (Invitrogen). Proteins were detected with a SARS-CoV-2 Spike S2 rabbit polyclonal antibody (Spike S2 PAb) (https://www.sinobiological.com/antibodies/cov-spike-40590-t62) (1:2000 dilution), followed by anti-rabbit IgG secondary antibody, each for 1 h at RT. β-actin was included as a loading control. $M_r$, relative molecular weight (K denotes x 1,000).

## Detection of CoV spike binding to soluble *Hs*ACE2 by flow cytometry

293T cells were seeded in a 10-cm dish. 24 h later, cells were transfected with 5 μg of a plasmid encoding an expression vector and the spike proteins of WIV1-CoV, Rs3367-CoV, or SHC014-CoV (WT or Y623H). At 24 h post-transfection, cells were harvested and incubated with 5 μg/mL trypsin (Gibco, Cat# 15050057) or PBS for 1 h at 4°C. Trypsin was subsequently stopped with 10% FBS for 15 min at 4°C. Cells were then incubated with 50 nM soluble *Hs*ACE2 for 1 h at 4°C and stained with Strep-Tactin XT PE (IBA Lifesciences, Cat# 6-5400-001) for 1 h at 4°C. After washing, stained cells were filtered through a 41-μm nylon net filter (Millipore) and analyzed using a Cytek Aurora Flow Cytometer and FlowJo software.

## ELISA for scVSV-SHC014-CoV spike binding to soluble *Hs*ACE2

High-protein binding half-area 96-well plates (Corning) were coated with 2 μg/mL of soluble *Hs*ACE2 overnight at 4°C. Plates were washed with PBS and blocked with 3% BSA in PBS for 1 h at 37°C. Serial 3-fold dilutions of pre-titrated amounts of scVSV-SHC014-CoV, scVSV-WIV1-CoV, scVSV-Rs3367-CoV, or chimeric constructs were added to blocked plates for 1 h at 37°C. Plates were then washed with PBS and incubated with 10 nM of ADG-2 for 1 h at 37°C. Plates were washed and goat anti-human IgG secondary antibody conjugated to horseradish peroxidase (HRP) was added and incubated for 1 h at 37°C. Plates were washed, and 1-Step Ultra TMB-ELISA Substrate Solution (Thermo Fisher) was then added and quenched with 0.5 M $H_2SO_4$. Absorbance was measured at 450 nm.

## Viral infectivity assays

DBT-9 cells overexpressing *Rs*ACE2 or human ACE2 were seeded on 96-well plates at 1.8 x $10^4$ cells per well. 24 h after seeding, 3-fold serial dilutions of genome-normalized amounts of scVSV particles wewre added to cells for 12–18 h at 37°C. For experiments including $NH_4Cl$, cells were pretreated with $NH_4Cl$ for 20 min at 37°C. Virus was diluted in media containing 20nM $NH_4Cl$ and then incubated on cells for 12–18 hours at 37°C. Cells were fixed with 4% paraformaldehyde and stained with Hoechst-33342 (Invitrogen). eGFP+ cells were counted using a Cytation 5 reader (BioTek Instruments). Infection levels were determined as the percentage of eGFP+ cells over the total number of cells, enumerated by Hoechst staining.

## Trypsin infectivity assays

Vero cells or Vero81-ACE2 knockout cells stably transfected with empty vector or *Rs*ACE2 were seeded on 96-well plates at 1.5 x $10^4$ cells per well. At 24 h after seeding, pre-diluted amounts of scVSV-WIV1-CoV, scVSV-Rs3367-CoV, scVSV-SHC014-CoV (WT or Y623H) in high-glucose DMEM without FBS were mixed at 1:1 with trypsin (Gibco, Cat# 15050057) to a final concentration of 200 μg/mL on ice. Media was removed from the cells, and the wells were washed with PBS. Virus:trypsin mixture was then added to cells. Plates were centrifuged at 1200 xg for 1 h at 4°C, followed by 10% FBS to inhibit trypsin. Cells were incubated at 37°C for 12–18 h, then fixed with 4% paraformaldehyde, washed with PBS and stained with Hoechst-33342. Infection levels were determined as the percentage of eGFP+ cells over the total number of cells.

## ADG-2 neutralization assay

Pre-diluted amounts of scVSV-SHC014-CoV, scVSV-Rs3367-CoV, scVSV-WIV1-CoV, or chimeric constructs of each spike were mixed with serial dilutions of mAb ADG-2 for 1 h at 37°C. Virus/antibody mixture was added to DBT-9 cells overexpressing *Ra*ACE2 cells, pre-seeded in 96 well plates. Cells were incubated at 37°C for 12–18 h and then fixed with 4% paraformaldehyde, washed with PBS and stained with Hoechst-33342 (Invitrogen). eGFP+ cells were counted using a Cytation 5 reader (BioTek Instruments). Infection levels were determined as the percentage of eGFP+ cells over the total number of cells. Infection was normalized to no-mAb controls.

## AlphaFold2 modeling

AlphaFold2 (alphafold2_multimer_v3) was executed via ColabFold v1.5.5 [66,67] with the Rs3367-CoV spike sequence as input. The cryo-EM structure of the RsSHC014-CoV spike (PDB ID: 8WLU [11]) was used as a template for structure prediction. Default settings were used for all other options.

## Betacoronavirus spike sequence dataset assembly

We downloaded amino acid sequences for all proteins (174,493,551) belonging to any betacoronavirus (9,201,392) annotated in the BV-BRC database [68]. Then, we assembled a local blast database using makeblastdb and blasted the amino acid sequence of the spike protein from WIV1-CoV against it [69,70]. We obtained 12,993,256 hits with a minimum e-value < 3x10$^{-3}$. Following this, we filtered out the initial hits, keeping only those hits that: 1) were reported with the lowest e-value per genome (9,131,029 hits); 2) belonged to sarbecoviruses (9,127,450 hits); 3) aligned with > 80% of the query sequence (7,265,072 hits) and; 4) do not contain unknown residues in their sequence (5,944,141 hits). Finally, we assembled a non-redundant dataset of 232,303 amino acid sequences representing the sequence space of the spike protein in 5,944,141 sarbecoviruses using CD-hit (https://github.com/weizhongli/cdhit/) [71] at 100% sequence identity (cd-hit -c 1 -n 5 -aL 0.9) [71].

## Identification of residue identities at particular positions

We ran non-self pairwise sequence alignments between the reference query sequence (the spike protein from Rs3367-CoV) and all other 232,302 representative amino acid sequences. Pairwise sequence alignments were parsed to identify the residue identities at position 623 (sequence numbering corresponding to Rs3367-CoV) using the Biopython package [72].

## Statistical analysis

Statistical details for each experiment can be found in respective figure legends, including statistical test used and number of replicates (n). Statistical analyses were carried out in GraphPad Prism.

## Supporting information

**S1 Fig. WIV1-CoV and Rs3367-CoV spikes have no difference in cell surface expression level. (a)** Schematic representation of the amino acid differences between WIV1-CoV and SHC014-CoV in three regions of the spike proteins. Differences are highlighted in red. **(b-d)** 293T cells were transfected with plasmids expressing either WIV1-CoV or Rs3367-CoV spike and harvested at 24 h post-transfection. **(b-c)** Cells were immunostained for cell surface expression by S309, a S1-directed mAb, followed by a fluorescent secondary antibody, and analyzed using flow cytometry. Single cells were gated based on SSC-A and FSC-A and AF488-positive cells were selected by gating on transfected cells that were only stained with the fluorescent secondary antibody for background (average±SD, n = 5 from 3 independent experiments). Representative histogram shown **(c)**. **(d)** Transfected cells were lysed and spike expression levels were analyzed by western blot using a S2-directed polyclonal antibody. β-actin was included as a loading control. **(e-f)** Genome-normalized amounts of scVSVs were diluted with 3-fold dilutions onto ELISA plates, followed by a spike-specific mAb S309 and incubation with an anti-human HRP-conjugated secondary antibody (average±SD, n = 6 from 2 independent experiments). A range of $1.0 \times 10^7$ to $1.37 \times 10^4$ viral GEQ was used. WIV1-CoV vs. Rs3367-CoV were compared with a non-parametric Mann-Whitney test, ns p > 0.05; ** p < 0.01; *** p < 0.001; **** p < 0.0001.
(TIF)

**S2 Fig. Spike amino acid substitution Y623H causes changes in RBD-up availability between WIV1-CoV and SHC014-CoV. (a)** Schematic representation of the CoV S protein. Locations and amino acid identities of the two point

mutations that differ between Rs3367-CoV and WIV1-CoV are highlighted. **(b-c)** Genome-normalized amounts of scVSVs bearing spikes of WIV1-CoV (WT, K85R, SHC014 RBD, or H623Y) **(b)**, or SHC014-CoV (WT or Y623H) **(c)** were diluted with 3-fold dilutions onto ELISA plates precoated with soluble *Hs*ACE2, followed by a spike-specific mAb, and incubation with an anti-human HRP-conjugated secondary antibody (average±SD, n=6–8 from 3-4 independent experiments). A range of $9 \times 10^7$ to $3.7 \times 10^5$ viral GEQ was used. Groups were compared against WT with two-way ANOVA with Tukey's correction for multiple comparisons, ns $p > 0.05$; ** $p < 0.01$; *** $p < 0.001$; **** $p < 0.0001$. **(d-e)** Pre-titrated amounts of scVSVs bearing spikes of WT or chimeric WIV1-CoV **(d)** or SHC014-CoV **(e)** were incubated with serial 3-fold dilutions of ADG-2 mAb, starting at 100 nM, for 1 h at 37°C. Virus:ADG-2 mixtures were applied to monolayers of DBT-9 cells overexpressing *Rs*ACE2. At 16–18 hours post-infection, infected cells were scored by eGFP expression (average±SD, n=9 from 3 independent experiments). Relative infectivity (%) was calculated by normalizing to no-mAb controls for each virus. AUC values were calculated for each curve, and groups were compared by one-way ANOVA with Dunnett's post hoc test, ns $p > 0.05$; ** $p < 0.01$; *** $p < 0.001$; **** $p < 0.0001$. Only the statistically significant comparisons are shown.
(TIF)

**S3 Fig. Amino acid Y623 in Rs3367-CoV is highly conserved amongst sarbecoviruses except for WIV1-CoV.**
Alignment of amino acid spike sequences surrounding position 623 of the Rs3367-CoV spike protein (blue rectangle) for a panel of sarbecoviruses. Viruses are color-coded by clade; 1a: SARS-CoV-like (red), 1b: SARS-CoV-2-like (green), 2: Southeast Asian bat-origin CoV (blue), 3: non-Asian bat-origin CoV (purple). A larger sequence-based analysis using the spike of 5,944,141 sarbecoviruses was performed to assess amino acid conservation at position 623 (S1 File).
(TIF)

**S4 Fig. Flow cytometry plots for ADG-2 and HsACE2 binding by spike-transfected cells.** Flow cytometry analysis of 293T cells transfected with plasmids expressing either WIV1-CoV or Rs3367-CoV spikes. Cells were immunostained for cell surface expression by the S1-directed mAb ADG-2 followed by a fluorescent secondary antibody **(a)**, or soluble *Hs*ACE2 followed by Streptactin PE **(b)** and analyzed using flow cytometry. Representative histograms are shown.
(TIF)

**S5 Fig. Trypsin treatment cleaves bat CoV spike proteins. (a-b)** 293T cells were transfected with plasmids to express the indicated spike proteins. At 24 h post-transfection, cells were harvested, lysed, and spike protein expression was analyzed by western blot with a CoV S2-directed polyclonal antibody. Bands corresponding to uncleaved spike (S0) or the cleaved S2 subunit (S2) are indicated based on their relative molecular weights ($M_r$). β-actin was included as a loading control. **(b)** At 24 h post-transfection, spike-expressing cells were incubated with increasing amounts of trypsin for 1 h at 37°C.
(TIF)

**S6 Fig. Trypsin treatment increases ACE2-dependent cell entry and ACE2 binding by bat CoV spikes. scVSVs bearing the spikes of SHC014-CoV (WT or Y623H) were mixed with 200 µg/mL of trypsin and added onto Vero cells.** At 16–18 h post-infection, infection levels were scored by eGFP expression (average±SD. n=6 from 3 independent experiments). Infectivity values were normalized to no-trypsin controls. Groups (no trypsin vs. trypsin treatment) were compared with unpaired t-test with Welch's correction, ns $p > 0.05$; ** $p < 0.01$; *** $p < 0.001$; **** $p < 0.0001$.
(TIF)

**S7 Fig. Flow cytometry plots for trypsin-treated spike-transfected cells.** Flow cytometry analysis of 293T cells transfected with plasmids expressing the spike proteins of either **(a)** WIV1-CoV or **(b)** Rs3367-CoV. Cells were harvested and treated with 5 µg/mL of trypsin for 1 h. Cells were incubated with soluble *Hs*ACE2 followed by Streptactin PE and analyzed using flow cytometry. Representative histograms are shown.
(TIF)

**S8 Fig. Gating strategy for flow cytometry analysis of binding by spike-transfected cells.** Representative gating strategy for analysis of 293T cells transfected with plasmids expressing WIV1-CoV or Rs3367-CoV spike proteins. Cells were gated based on SSC-A and FSC-A and single cells were further gated on FSC-A and FSC-H. Cells were stained with either **(a)** ADG-2 followed by an anti-human Alexa Fluor 488 antibody, **(b)** HsACE2 followed by Streptactin PE, or **(c)** S309 followed by anti-human Alexa Fluor 488 antibody. The gate for positive cells was set based on a negative cell population transfected with a control vector and stained as above.
(TIF)

**S9 Fig. Mutational hotspot in the S2 FPPR region associated with rescue of rVSVs bearing Rs3367-CoV and SHC014-CoV spikes. (a)** Overview of an Alphafold2 model of the Rs3367-CoV spike trimer. The boxed region corresponds to the fusion-peptide proximal region (FPPR). **(b-c)** FPPR in the spikes of Rs3367-CoV **(b)** and SHC014-CoV (PDB: 8WLU) **(c)**. Residues in Rs3367-CoV (this study; Q836, N943, V946) and SHC014-CoV (A835) [12] at which substitutions arose during rVSV rescue are highlighted, together with interacting residues.
(TIF)

**S10 Fig. DBT9-HsACE2 cells were pretreated with 20 mM NH$_4$Cl or DMEM for 20 min prior to addition of genome-normalized scVSV particles bearing Rs3367-CoV(WT) or Rs3367-CoV(Y623H) spikes in media with or without 20 mM NH$_4$Cl.** Infection was scored by eGFP expression at 16–18 hours post-infection (average±SD. n = 4 from 2 independent experiments).
(TIF)

**S1 File. Sarbecovirus S proteins with non-conserved residues at position 623.** Analysis of a non-redundant dataset of 232,303 amino acid sequences representing the sequence space of the spike protein in 5,944,141 sarbecoviruses to identify the residue identities at position 623 (sequence numbering corresponding to Rs3367-CoV).
(XLSX)

**S2 File. Reference nucleotide sequence file of codon-optimized Rs3367-CoV spike encoding a 21-amino acid deletion at the C-terminus.** Nucleotide sequence cloned into a VSV genome vector to generate rVSV-Rs3367-CoV and used as a reference for aligning long-read sequences (see S3 File below).
(FASTA)

**S3 File. Variant calls in the spike gene from the sequencing of viral cDNA derived from two rVSV-Rs3367-CoV populations.** Analysis of spike sequences derived from long-read sequencing of cDNA isolated from viral populations generated by the plasmid-based rVSV rescue of rVSV-Rs3367-CoV. See Materials and Methods for the analysis workflow. Variant calls are reported in VCF format and were filtered for a quality score (QUAL) ⩾ 50. Data from two biological replicates are shown.
(XLSX)

## Acknowledgments

We thank J. Janer, K. Paez, and M. Ramirez for laboratory management and technical support. We thank J.N. Catanzaro, Jr. and R.S. Baric for their provision of Vero81- and DBT-9–derived cell lines. We acknowledge the Einstein Flow Cytometry Core Facility (supported by grant P30CA013330 from the U.S. National Cancer Institute) for flow cytometry support.

## Author contributions

**Conceptualization:** Alexandra L. Tse, Kartik Chandran, Emily Happy Miller.

**Investigation:** Alexandra L. Tse, Gorka Lasso, Jacob Berrigan, Jason S. McLellan, Kartik Chandran, Emily Happy Miller.

**Methodology:** Alexandra L. Tse, Gorka Lasso, Jacob Berrigan, Jason S. McLellan, Kartik Chandran, Emily Happy Miller.

**Supervision:** Jason S. McLellan, Kartik Chandran, Emily Happy Miller.

**Visualization:** Alexandra L. Tse, Gorka Lasso, Jason S. McLellan, Kartik Chandran, Emily Happy Miller.

**Writing – original draft:** Alexandra L. Tse, Kartik Chandran, Emily Happy Miller.

**Writing – review & editing:** Alexandra L. Tse, Gorka Lasso, Jacob Berrigan, Jason S. McLellan, Kartik Chandran, Emily Happy Miller.

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
