## [Decision Letter · Decision Letter 0]

2 Jun 2025

Bat sarbecovirus WIV1-CoV bears an adaptive mutation that alters spike dynamics and enhances ACE2 binding

PLOS Pathogens

Dear Dr. Chandran,

Please pay particular attention to the suggestions and additional data requested by reviewer #1 and reviewer #3.

Additional data are needed to further support the conclusions. Spike protein expression levels and efficiency of virion incorporation can modulate entry efficiency and should be controlled for. Further, the conclusion that mutation Y623H is a cell culture adaptation needs to be supported by additional data – it should be examined whether chimeric VSV encoding rVSV- Rs3367-S acquires Y623H or related mutations upon passaging in cell culture. Finally, it should be examined whether trypsin-promoted entry of Rs3367-S bearing particles remains ACE2-dependent.

Please submit your revised manuscript within 60 days Aug 01 2025 11:59PM. If you will need more time than this to complete your revisions, please reply to this message or contact the journal office at plospathogens@plos.org. Please include the following items when submitting your revised manuscript:

We look forward to receiving your revised manuscript.

Kind regards,

Kevin K Ariën, Ph.D.

Academic Editor

PLOS Pathogens

Sonja Best

Section Editor

PLOS Pathogens

Editor-in-Chief

PLOS Pathogens

orcid.org/0000-0003-2946-9497

Editor-in-Chief

PLOS Pathogens

orcid.org/0000-0002-7699-2064

**Journal Requirements:**

3) We notice that your supplementary Figures are included in the manuscript file. Please remove them and upload them with the file type 'Supporting Information'. Please ensure that each Supporting Information file has a legend listed in the manuscript after the references list.

4) We note that your Data Availability Statement is currently as follows: "All data associated with this study are present in the paper and its Supplementary files.". Please confirm at this time whether or not your submission contains all raw data required to replicate the results of your study. Authors must share the “minimal data set” for their submission. PLOS defines the minimal data set to consist of the data required to replicate all study findings reported in the article, as well as related metadata and methods (https://journals.plos.org/plosone/s/data-availability#loc-minimal-data-set-definition).

1) If the funders had no role in your study, please state: "The funders had no role in study design, data collection and analysis, decision to publish, or preparation of the manuscript."

2) If any authors received a salary from any of your funders, please state which authors and which funders.

6) Please ensure that the funders and grant numbers match between the Financial Disclosure field and the Funding Information tab in your submission form. Note that the funders must be provided in the same order in both places as well. Currently, the order of this grant "F-0003-1962060" is different in both places.

7) Please provide a completed 'Competing Interests' statement, including any COIs declared by your co-authors. If you have no competing interests to declare, please state "The authors have declared that no competing interests exist". Otherwise please declare all competing interests beginning with the statement "I have read the journal's policy and the authors of this manuscript have the following competing interests:"

**Reviewers' Comments:**

Reviewer's Responses to Questions

**Part I - Summary**

Reviewer #1: Tse and colleagues investigated determinants of sarbecovirus spike proteins that allow for efficient usage of ACE2. In brief, they provide evidence that mutation Y623H increases the open conformation of the S protein as well as ACE2 binding and cellular entry. Further, it is suggested that Y623H can be acquired upon cell culture adaptation of sarbecovirus. Collectively, the findings suggest that not only acquisition of a multibasic cleavage site but also acquisition of S protein mutations that favor the open conformation of the S protein and increase ACE2 binding could increase the zoonotic potential for sarbecoviruses. These findings are of interest but additional data are needed to confirm the conclusions drawn.

Reviewer #2: The manuscript titled “Bat sarbecovirus WIV1-CoV bears an adaptive mutation that alters spike dynamics and enhances ACE2 binding” by Tse A.L. et al. provides important insights into the potential mechanisms by which WIV1-CoV may have adapted for infection of human cells. The authors investigate the role of the Y623H substitution in the spike (S) protein, which appears to enhance infectivity in human cells compared to the closely related Rs3367-CoV. This substitution is proposed to alter interactions between the 630 loop, the CTD2 β-sheet, and the NTD-RBD linker, thereby increasing RBD mobility, enhancing RBD availability, and improving ACE2 binding.

Given that only two amino acid differences distinguish WIV1-CoV from Rs3367-CoV—including Y623H—the authors suggest that WIV1-CoV likely arose from Rs3367-CoV via adaptation during early passage in human cells. This mechanism of adaptation highlights an alternative zoonotic spillover pathway, in which substitutions that facilitate spike opening and receptor engagement may precede—or potentially obviate—the need for substitutions enhancing spike cleavage. These early adaptive changes could serve as useful prognostic markers for zoonotic risk.

The manuscript is well written, with clearly presented information and strong experimental support for the main conclusions.

Reviewer #3: The manuscript characterizes an amino acid substitution (Y623H) in WIV1 bat coronavirus that appears to have been selected for by virus isolation in Vero E6 cells and confers increased ACE2 binding and entry capacity to WIV1 and other very closely-related bat coronavirus spikes. Novelty and impact of the investigation are somewhat diminished by the recent publication from Qiao and Wang (PMID: 39028202) describing effect of the Y623H substitution on spike-mediated entry in the WIV1 and SHC014 spikes. The current study supports that finding, but also contributes important additional information indicating that this substitution was likely selected during viral outgrowth. The study also provides additional findings on relative ACE2 binding, experimental evidence for the “open” spike conformation with increased RBD availability conferred by Y623H, and effect of trypsin cleavage on entry of mutants and wild-type. While there are several ways the study should be improved, the experiments are generally rigorous and conclusions well-supported. The manuscript is very well written and easy to follow. With several improvements, these findings may contribute to our understanding of a specific mechanism by which native bat coronaviruses may evolve to efficiently enter human cells.

**Part II – Major Issues: Key Experiments Required for Acceptance**

Reviewer #1: Major

While the Y623H change markedly increases SHC014-CoV S protein-driven entry, entry activity is not restored to the same level measured for WIV-1 S protein. Therefore, it is important to analyze whether introducing NTD/RBD of WIV-1 into SHC014-CoV S increases entry efficiency. Expression of all chimeras (figure 1) in transfected cells or, preferably, incorporation into particles needs to be determined in order to allow for solid conclusions.

The conclusion that Y623H is a cell culture adaptation must be bolstered by functional data, considering that the mutation might also have been acquired during circulation in bats. Thus, it should be analyzed whether rVSV- Rs3367-S acquires Y623H or related changes upon passaging in cell culture.

The results shown in figure 3 must be bolstered by data on S protein incorporation into particles.

Treatment of sarbecovirus spike proteins with trypsin frequently augments infectivity but can also allow for ACE2-independent entry. Was entry driven by trypsin-treated WIV-1 S and Rs3367-S ACE2-dependent?

Reviewer #2: (No Response)

Reviewer #3: 1. In this VSV-based pseudovirus infection system with ACE2-expressing DBT-9 cells, does virus entry occur via the cell surface pathway, the endocytic pathway, or both? Does the Y623H substitution drive preferential utilization of one pathway compared to the other? This may be key to understanding the mechanism of altered entry and is brought up briefly in discussion but not investigated. Experimental determination of this would contribute greatly to study novelty and impact relative to previously published work.

2. While the manuscript clearly demonstrates that spike variants do not differ in expression in plasmid-transfected cells, what about incorporation of spike into VSV particles? It would be more convincing to test the level of spike present in purified virions, such as Western blot on sucrose gradient-purified pseudovirus (standardized by genome equivalents).

3. Conclusions are made regarding adaptation of a Rs3367-like CoV to human and primate (Vero E6) cell culture, but cell entry is tested using only bat host (Rhinolophus affinis) ACE2. Would results differ if human or primate ACE2 were tested? Why test RaACE2 for entry and human ACE2 for binding? Could amino acid differences between ACE2 derived from these species affect the results?

**Part III – Minor Issues: Editorial and Data Presentation Modifications**

Reviewer #1: It should be indicated whether lanes from the same immunoblot are shown in supplemental figure 1D or whether the results of different gels are shown. In the latter case, conclusions on on relative expression levels cannot be drawn.

Reviewer #2: 1-Information on viral stock titers is missing. How were GEQ/mL values calculated?

2-Figure 3B: The x-axis labels are unclear, especially in the context of a starting concentration of 2.5 µM and a 3-fold serial dilution. Please clarify the actual concentrations of ADG-2 used.

3-The AlphaFold2 modeling and the corresponding Figure 6 are not described in the Results or Methods sections and are mentioned only in the Discussion. These should be integrated more appropriately into the main text.

Reviewer #3: 1. Is it known whether DBT-9 murine cells express TMPRSS2 or other proteases that may impact entry capacity?

2. Fig. 1 title: “Y623H amino acid substitution causes binding and infectivity differences…” This data indicates infectivity differences but does not specifically address binding.

3. The AlphaFold spike modelling (Fig 6) constitutes an investigation that produced findings and should be described in the Results section, rather than solely the Discussion.

4. In methods, define “HsACE2” as Homo sapiens ACE2.

PLOS authors have the option to publish the peer review history of their article (what does this mean? ). If published, this will include your full peer review and any attached files.

**Do you want your identity to be public for this peer review?** For information about this choice, including consent withdrawal, please see our Privacy Policy .

Reviewer #1: No

Reviewer #2: No

Reviewer #3: No

**Figure resubmission:**

**Reproducibility:**



---

## [Decision Letter · Decision Letter 1]

7 Oct 2025

Dear Dr. Chandran,

We are pleased to inform you that your manuscript 'Bat sarbecovirus WIV1-CoV bears an adaptive mutation that alters spike dynamics and enhances ACE2 binding' has been provisionally accepted for publication in PLOS Pathogens.

Best regards,

Kevin K Ariën, Ph.D.

Academic Editor

PLOS Pathogens

Sonja Best

Section Editor

PLOS Pathogens

Sumita Bhaduri-McIntosh

Editor-in-Chief

PLOS Pathogens

orcid.org/0000-0003-2946-9497

Michael Malim

Editor-in-Chief

PLOS Pathogens

orcid.org/0000-0002-7699-2064

Reviewer Comments (if any, and for reference):

Reviewer's Responses to Questions

**Part I - Summary**

Reviewer #2: (No Response)

Reviewer #3: In my view, the authors have thoroughly addressed all major issues brought forth by both reviewers, resulting in a substantially improved manuscript with potential for high impact in the field.

**Part II – Major Issues: Key Experiments Required for Acceptance**

Reviewer #2: (No Response)

Reviewer #3: (No Response)

**Part III – Minor Issues: Editorial and Data Presentation Modifications**

Reviewer #2: (No Response)

Reviewer #3: (No Response)

PLOS authors have the option to publish the peer review history of their article (what does this mean? ). If published, this will include your full peer review and any attached files.

**Do you want your identity to be public for this peer review?** For information about this choice, including consent withdrawal, please see our Privacy Policy .

Reviewer #2: No

Reviewer #3: No

---

## [Editor Report · Acceptance letter]

Dear Dr. Chandran,

We are delighted to inform you that your manuscript, "Bat sarbecovirus WIV1-CoV bears an adaptive mutation that alters spike dynamics and enhances ACE2 binding," has been formally accepted for publication in PLOS Pathogens.

Best regards,

Sumita Bhaduri-McIntosh

Editor-in-Chief

PLOS Pathogens

orcid.org/0000-0003-2946-9497

Michael Malim

Editor-in-Chief

PLOS Pathogens

orcid.org/0000-0002-7699-2064